# In Silico Identification of Lead Compounds for Pseudomonas Aeruginosa PqsA Enzyme: Computational Study to Block Biofilm Formation

**DOI:** 10.3390/biomedicines11030961

**Published:** 2023-03-21

**Authors:** Muhammad Shahab, Muhammad Danial, Taimur Khan, Chaoqun Liang, Xiuyuan Duan, Daixi Wang, Hanzi Gao, Guojun Zheng

**Affiliations:** 1State Key Laboratories of Chemical Resources Engineering, Beijing University of Chemical Technology, Beijing 100029, China; 2Shenzhen Institute of Advanced Technology, University of Chinese Academy of Sciences, Shenzhen 518055, China

**Keywords:** pqsA gene, pharmacophore-based virtual screening, ZINC, Cambridge, MD simulation, biofilms

## Abstract

*Pseudomonas aeruginosa* is an opportunistic Gram-negative bacterium implicated in acute and chronic nosocomial infections and a leading cause of patient mortality. *Pseudomonas aeruginosa* infections are frequently associated with the development of biofilms, which give the bacteria additional drug resistance and increase their virulence. The goal of this study was to find strong compounds that block the Anthranilate-CoA ligase enzyme made by the pqsA gene. This would stop the *P. aeruginosa* quorum signaling system. This enzyme plays a crucial role in the pathogenicity of *P. aeruginosa* by producing autoinducers for cell-to-cell communication that lead to the production of biofilms. Pharmacophore-based virtual screening was carried out utilizing a library of commercially accessible enzyme inhibitors. The most promising hits obtained during virtual screening were put through molecular docking with the help of MOE. The virtual screening yielded 7/160 and 10/249 hits (ZINC and Chembridge). Finally, 2/7 ZINC hits and 2/10 ChemBridge hits were selected as potent lead compounds employing diverse scaffolds due to their high pqsA enzyme binding affinity. The results of the pharmacophore-based virtual screening were subsequently verified using a molecular dynamic simulation-based study (MDS). Using MDS and post-MDS, the stability of the complexes was evaluated. The most promising lead compounds exhibited a high binding affinity towards protein-binding pocket and interacted with the catalytic dyad. At least one of the scaffolds selected will possibly prove useful for future research. However, further scientific confirmation in the form of preclinical and clinical research is required before implementation.

## 1. Introduction

*Pseudomonas aeruginosa* is a type of Gram-negative bacterium that can cause infections in humans, particularly in healthcare settings, and is able to thrive in a variety of environments and conditions. This is due to its genome plasticity, resistance to environmental stresses, great metabolic versatility, high resistance to antibiotics, powerful biofilm-forming ability, and, very importantly, the expression of quorum sensing-regulated virulence factors [1]. According to the WHO, *P. aeruginosa* is a high-priority disease, and there is a pressing need for new treatments [2]. Multi-drug resistant *P. aeruginosa* strains are a major issue for hospitals [3]. *P. aeruginosa* is a leading pathogen among immunocompromised patients, particularly those with diseases such as HIV, diffused panbronchiolitis, cystic fibrosis, and chronic obstructive pulmonary disease, and cancer patients receiving chemotherapy [4,5]. It is also one of the main causes of nosocomial infections, accounting for almost 10% of such infections [6]. Cystic fibrosis patients primarily die from chronic infections and lung inflammation caused by *P. aeruginosa* [7]. Multidrug-resistant *P. aeruginosa* strains have a negative impact on patient outcomes, including increased mortality, hospital visits, length of stay, and cost [8,9].

*P. aeruginosa* is a member of the ESKAPE panel of pathogens, which are multidrug-resistant bacteria known as “superbugs” [10]. It is recognized by the World Health Organization as one of the most important priority infections due to its resistance to a wide range of antibiotics, including third-generation cephalosporins and carbapenems [11]. The ability of bacteria to form biofilms also makes them more resistant to antibiotics, with *P. aeruginosa* strains that produce biofilms showing resistance to fluoroquinolones and gentamicin by 20–30% and 12–22%, respectively [12]. Quorum sensing (QS) regulates biofilm development in various bacteria and is a key intercellular signaling mechanism for virulence factor production, genetic competence, antimicrobial peptide production, fruiting body formation, plasmid conjugation, and symbiosis [13,14]. Quorum sensing is a process by which bacteria communicate with each other by secreting and detecting small, diffusible signal molecules called autoinducers (AIs) [15]. A wide range of processes are regulated by AIs, including the release of virulence factors, swimming motility, the production of secondary metabolites, the development of biofilms, and antibiotic resistance [16].

Interfering with the way bacteria communicate with one another, known as quorum sensing -mediated signaling, is a promising strategy for limiting the proliferation of harmful pathogens [17,18,19,20]. By disrupting this process, it becomes possible to reduce the pathogenicity of the bacteria and make them more susceptible to removal by the host’s immune system. The use of quorum sensing inhibitors is being researched as a preventative measure to manage pathogens that are resistant to antibiotics, which are becoming an increasingly significant public health concern. *P. aeruginosa* has three main quorum sensing systems: rhl, las, and pqs. These systems control how virulence proteins are made and how cells talk to each other. As shown in Figure 1, the las system starts the expression of AI receptors, which positively affects both the rhl and pqs systems (PqsR and RhlR). The RhlR and PqsR are also activated by the binding of their respective AIs [21,22].

The pqs system, in particular, uses two signal molecules: Pseudomonas quinolone signal, also known as 2,3,5-trihydroxy-4 (1H)-quinoline (PQS), and 2-heptyl-4- quinoline (HHQ). Attachment of these molecules to the PqsR activates many genes that cause biofilm formation [12]. This system plays a crucial role in the virulence of *P. aeruginosa*, as it regulates the production of several virulence factors and the formation of biofilms, which are protective structures that allow the bacteria to evade the host’s immune system. Inhibition of the enzyme PqsA affects the synthesis of PQS signal molecules, gene regulation controlled by PqsR, and biofilm formation.

In this study, we employed cutting-edge in silico drug discovery technology to predict potent and novel inhibitors of quorum sensing (QS). The process of discovering new drugs is a complex and time-consuming endeavor that typically requires significant resources, including years of research and billions of dollars in funding [23]. However, the use of advanced computational approaches such as molecular docking simulations and virtual screening has greatly facilitated the drug discovery process by allowing researchers to identify new lead compounds against specific targets through rational, principle-based approaches based on theoretical chemistry. These computational methods have saved researchers time and resources [24,25]. Computational methods, such as virtual screening, docking, and binding free energy analysis, can help identify potential drug candidates from compound libraries, saving time and money in the drug development process [26].

## 2. Methodology

The overall mechanism and various tools used in this study for designing lead compounds by rational drug design are depicted in Figure 2.

### 2.1. Retrieval of Protein

The 3D structure of the protein is required to comprehend the molecular interaction study of proteins with ligands. For this purpose, the initial crystallographic structure of the target protein (PDB ID: 5OE3) (accessed on: 1 January 2023) and its co-crystallized ligand structure were extracted from the protein using MOE. First, the structure was checked for any missing chain breaks or missing atoms. It was then prepared by removing the water molecules and the non-protein atoms from the crystal structure. Hydrogens were added, and the bond order assigned according to the protocol “compute Quick preparation” in MOE 2020. All the atoms in the Amber-22 (Amber ff19SB) forcefield were used to refine the protein structure. Hydrogens were added to amino acids using 3D protonation, then energy was minimized with the MOE 2020 program. To date, only the N-terminal domain is available in the protein databank; however, understanding the dynamic features of the full protein upon binding is important [27,28]. We screened potential inhibitors using the N-terminal domain structure against commercially available datasets (ZINC and Chembridge) (accessed on: 1 January 2023)) since the C-terminal region did not affect the binding pocket of the N-terminal domain.

### 2.2. Preparation of Quorum Sensing Inhibitor

Crystal structures of the N-terminal domain of anthranilate-CoA ligase PqsA, the first enzyme of PQS biosynthesis, in complex with 6-fluoroanthraniloyl-AMP (6FABA-AMP) at 1.7 Å resolution were selected from the protein databank (https://www.rcsb.org/structure/5OE3 (accessed on: 1 January 2023)) and used as a reference drugs in this study. The ligands were deposited to MOE for pre-processing, including protonation, ionization, specified counter ions, and energy minimization using AMBER force field ff19SB.

### 2.3. Pharmacophore Based Virtual Screening

To find the lead compound against PqsA (key enzyme in *P. aeruginosa* quinolone signaling), pharmacophore-based virtual screenings were created based on the protein complex with the ligand. The selected protein was analyzed using MOE software for H-bond donors/acceptors, hydrophilicity, lipophilic features and ionizable charges. The pharmacophore represents steric and electronic features for optimal interaction with a target and blocking its response. The common feature pharmacophore model was generated by using co-crystallized inhibitor 6-fluoroanthraniloyl-AMP. The model was then validated by two methods; first, a set of 8 active compounds were taken from the literature study and then screened against the generated pharmacophore model; second, the validation of the pharmacophore model was also carried out by examining its interaction with important amino acids in the receptor protein’s active pocket, based on the important chemical features of the pharmacophore. This was used to assess the accuracy of our predicted model through protein–ligand interactions.

### 2.4. Screening of Commercially Available Databases

In pharmacophore-based drug discovery, identifying novel and active molecules that are structurally similar is crucial. To achieve this, virtual screening was performed using the prepared pharmacophore model to identify potential lead compounds. In this connection, the ChemBridge [29] and ZINC [30], database were utilized for pharmacophore-based screening. The compounds most closely matching the selected pharmacophore features were then selected. By using MOE software against ZINC and Chembridge databases, the pharmacophore-based virtual screening was carried out [31]. From the screening, structurally diverse hits presenting a better fit to the generated pharmacophore model were recovered. Compounds with larger molecular weights > 500 KD; H-bond donors > 5; H-bond acceptors > 10; and Logp o/w > 5 were not selected. The final best hits were then examined for molecular docking and molecular dynamics simulation.

### 2.5. Molecular Docking

Molecular docking predicts small molecule–protein binding affinity to aid drug design [32]. The retrieved compound was placed into a three-dimensional grid representing the protein, and the position of the molecule was optimized to achieve a high binding energy. The 3D protonation of the target receptor was tracked using MOE software 2020 default parameters for optimal outcome. To refine the results, all compounds were docked into the PqsA binding site. The top 10 conformations from each hit were chosen for further study. The docking analysis underwent closer examination, putting more attention on protein/hit interactions and docking scores. To validate the molecular docking results, MDS was employed. All acquired hits were subjected to docking with the PqsA protein to determine the total number of interactions and leading compounds. The root mean square deviation (RMSD) between the co-crystallized and re-docked conformations for each ligand was calculated using the MOE SVL script and found to be 0.78 Å, indicating the reliability of the docking protocol. This was achieved by allowing 30 conformations using the default parameters in MOE: Triangle Matcher for placement, London dG and GBVI/WSA dG for rescoring, and Rigid Receptor for refinement [31]. Based on docking score and binding interaction, the top 4 compounds (from ZINC and Chembridge) were ranked. These 4 compounds displayed potency toward the target compared to the reference compound. The binding interactions and proteins were visualized using PyMOL. The top-scoring complexes were then run through a molecular dynamics simulation using Amber v.22 software.

### 2.6. Systematic Analysis of the Potent Lead Compound

Molecular dynamics simulation is a computational technique to study molecular behavior and interactions over time through mathematical models and offer rich information on the dynamics and structure of biomolecules, target proteins and drug interactions in therapy. The top four compounds with an improved docking score, binding affinity, energy, and interaction were analyzed through post trajectory analysis. The lead molecules were parameterized using GAFF and assigned ff19SB atom types with Antechamber, and their parameter files were generated using tLEaP [33]. All visualization was conducted in PyMol [34]. All-atom MD simulations and essential dynamics analysis were conducted in AMBER version 2022 [35]. The LEaP module was used to integrate hydrogen atoms into the crystal structure. Next, counter ions (Na^+^ and Cl^−^) were added to maintain the systems’ neutrality by using the tLEAP module. All systems were solvated in a TIP3P water model truncated octahedral box with a cut-off 12.0 Å buffer. The particle mesh Ewald (PME) method [36] was used to treat long-range electrostatic interactions. The SHAKE algorithm with a tolerance of 10^−5^ Å was applied to constrain all covalent bonds involving hydrogen atoms [37]. The CUDA-accelerated PMEMD was utilized for all MD simulations. The steepest descent method was applied to minimize the solvated systems with 10,000 steps, 800 ps heating phase, and 400 ps NVT equilibration. Temperature and pressure were regulated using Langevin’s algorithm with a time constant of 1.0 ps, isotropic scaling, and a relaxation time of 4.0 ps [38]. The analysis was conducted using CPPTRAJ implemented in Amber v 2022.

### 2.7. Hydrogen Bond Analysis

The Amber22 CPPTRAJ package is a tool that can be used to analyze molecular dynamics (MD) trajectories [39]. In order to comprehend the variations, it is important in determining the structural stability of PqsA enzyme. The total number of H-bonds play a crucial role in elucidation of the three-dimensional structure of the hits’ complexes. A total of 10,000 frames were taken during the MD simulation to assess the diversity among the ligand–protein complexes of the reference drug, and the identified hits was employed to examine the hydrogen bonds between the protein–ligand targets. Additionally, hydrogen bond analysis could also be used to investigate the binding of PqsA with the retrieved lead compound. H-bonding is described in this paper as occurring at a distance of 3.5 Å. All results were computed using the original program.

### 2.8. Dynamic Cross-Correlation Movement Analysis

The DCCM is a powerful tool that helps in understanding the dynamics of protein–ligand interactions. It helps to identify the key residues that are responsible for stability or instability of the complex. It also helps to identify the regions where the ligand can bind more efficiently and the regions where it can move away. This information is useful in designing new drugs or in understanding the mechanism of action of existing drugs. The DCCM graph shows positive and negative correlations, where positive correlations indicate ligand–protein movement in the same direction, stabilizing the system through interactions. Negative correlations imply instability of the complex or that the ligand has left the binding pocket, resulting in anti-parallel correlation. The color intensity of the DCCM map reflects the strength of the correlation, with blue to red representing positive and blue to light blue representing negative. The deeper the color, the stronger the correlation, and vice versa. These analyses display the correlation of amino acid residue movement over time and also evaluate the persistent correlations of domains [40].

The DCCM analysis was carried out by using Cα carbon atoms from 5000 snapshots. The Cα carbon atoms in the trajectories were cross-correlated with the displacements of backbone Cα atoms. The correlation coefficient between two atoms, i and j, is represented by Sij and is mathematically represented as:Sij = 〈Δri. Δrj〉/(〈Δri2〉〈Δri 2〉)1⁄2(1)

Here, the bracket “〈〉” defines time throughout the analysis, Δri or Δrj represent displacement vectors of ith or jth atoms with their average position, where Sij > 0 represents the movement with positive correlation (+1) between two atoms, i.e., atoms i and j, whereas when Sij < 0, it shows the movement with negative correlation (−1) between atoms i and j. Cpptraj was used for the analysis of DCCM, and Origin software was used for plotting the data.

### 2.9. Principal Component Analysis and Free Energy Landscape

The present study used the cpptraj package to conduct a PCA of the protein to identify the high-amplitude principal motions [41]. The dynamics behavior of all five systems was evaluated by calculating the covariance matrix based on Cartesian coordination of Cα atoms from 10,000 snapshots of the whole trajectories. By extracting eigenvectors and eigenvalues from the covariance matrix, the direction and mean square fluctuation of high-amplitude motions were determined. The first and last two principal components (PC1 and PC2) were plotted to monitor the motions of each system. The free energy landscape (FEL) was calculated using the first two principal components and equation:∆*G*(*X*) = −*KBT*lnP(*X*)(2)
where *X* represents the reaction coordinates, *KB* is the Boltzmann constant, and P(*X*) is the probability distribution of the system along the first two principal components. The FEL shows the folding and lowest energy stable states of the confirmation with minimal energy stable state, while the boundaries show intermediate conformations [42].

### 2.10. Binding Free Energy Calculations

The molecular mechanics generalized Born surface area (MMGBSA), computational method for predicting protein–ligand binding affinities by combining molecular mechanics and generalized Born implicit solvent models were applied [43]. Five hundred snapshots were taken from a 5 ns trajectory and analyzed using MM/GBSA (MMPBSA.py in Amber) to calculate binding free energy (ΔGbind) of simulated SMT proteins:ΔG_(bind)_ = ΔG_(R + L)_ − ΔG_(R)_ + ΔG_(L)…….1_(3)
G = E_(VDW)_ + E_(bond)_ + G_(GB)_ + E_(elec)_ + G_(SA)_ − TS_(S)…….2_(4)

The first equation above describes the calculation of binding free energy (ΔGbind) using the MM/GBSA method. The terms ΔG_(L)_, ΔG_(R)_, and ΔG_(R + L)_ represent the free energies of the ligand, receptor, and receptor–ligand complex, respectively. The second equation describes the components of the energy calculation, including dihedral energy (E_(bond)_), bond angles, van der Waals energy (ΔE_(VDW)_), and electrostatic energy (ΔE_(elec)_). The non-polar and polar contributions of solvation energy are represented as G_(SA)_ and G_(GB)_, respectively. The terms Ss and T refer to the solute entropy and the absolute temperature of the system [44,45].

## 3. Results and Discussion

### 3.1. Pharmacophore-based Virtual Screening

Pharmacophore modeling is a powerful technique that is used to identify and extract the key interactions between ligands and receptors. It is based on the principle that by schematically representing the essential components of molecular recognition, it is possible to represent and distinguish molecules that are likely to have similar biological activity and interactions with the target protein. Pharmacophore models describe 3D arrangements of functional groups involved in biological interactions with protein active sites. In ligand-based modeling, similar compounds are predicted to have similar biological activity and target protein binding. This is because the pharmacophore model focuses on the key features of the molecule that are involved in the interactions and binding, rather than the overall structure. Generally, ligand-based pharmacophore modeling is used to find new and potent ligands/inhibitors by comparing molecular similarity to known promising inhibitors, without protein structure information; this is a powerful advantage of this technique [46]. Pharmacophore modeling is widely used in the drug discovery process, as it allows for the efficient search and optimization of inhibitors. Using MOE software’s pharmacophore editor tool, a pharmacophore model was generated from known inhibitors (6-fluoroanthraniloyl-AMP), and seven important features selected with the pharmacophore query command. The model consisted of 2 Aromatic, 1 Acc, 1 Don, 1 Don & Acc, and 1 AtomQ features. These features are represented by different colors in the model, and they depict the interactions between the ligand and the receptor. The resulting model, as shown in Figure 3, is an essential characteristic of a pharmacophore model for the most active compound.

The pharmacophore model was validated by testing against a database of anti- quorum sensing drugs, including 6-fluoroanthraniloyl-AMP as a reference. It was then used to screen compounds from ZINC and ChemBridge libraries, resulting in the identification of numerous strong interacting compounds. A total of 445 and 1520 structurally diverse hits were retrieved from the ZINC and ChemBridge libraries, respectively, that fit the EHT pharmacophore model. Then, 160 and 249 hits were selected by applying Lipinski Ro5 (rules for predicting oral bioavailability) from the ZINC and ChemBridge libraries.

### 3.2. Molecular Docking

This method involves the placement of the small molecule into the binding pocket of the protein and the calculation of the energy of the complex. For the purpose of studying molecular interactions and selecting lead compounds, the chemical compounds that were identified using the pharmacophore model were docked into the binding site of a protein called PqsA chain. This process was performed using the molecular docking software Molecular operating environment (MOE 2020). The software generated 20 possible conformations per compound with default settings (Triangular Matcher placement, London dG and GBVI/WSA dG rescoring, Rigid Receptor refinement). To validate the accuracy of the docking, the RMSD was calculated between the co-crystallized and re-docked conformations using MOE’s SVL script, resulting in an RMSD of 0.78 Å, indicating a reliable protocol. Based on the docking score, the top four compounds were selected for further analysis. The potential anti-quorum sensing compounds that were discovered were seen to fit well within the PqsA drug target as shown in Figure 4. Four compounds with improved or equivalent binding strength and energy were selected for further analysis. The docking score, binding technique, pharmacophore mapping, stability of binding energy, binding affinity, and visual depiction of ligand interaction suggested that these lead compounds could be effective, diversified, and innovative protein medicines.

### 3.3. Binding of Selected Drug-like Compound

Molecular docking is a powerful tool for investigating how a ligand interacts with a drug target [47]. The results of molecular docking analysis indicated that the lead compound had better interaction with the PqsA enzyme than the reference compound. The ChemBridge54245649 was found to be the most active among the commercially available databases, with a docking score of −9.094. The predicted docking conformations showed that it may form eight hydrogen bond interactions with the target protein. This could make it a promising candidate for further drug development and testing, along with hydrophilic pie–pie interactions, salt-bridges and pie-stacking with the active residues, i.e., Phe 209, Gly 269, Thr 211, Thr 164, Arg 397, Asp 292, Asp382, Ile 301, Gly 302, Ala 303, Lys 172, Gly 269, Thr 211, and Pro 282 of the PqsA enzyme (Figure 5C). In the case of ChemBridge53910279, a total of four hydrogen bonds were established with docking scores of −8.533. Among the others with the pie–pie interactions, it could be seen that this compound blocked the key active side residues such as Lys 172, Gly 269, Ala 303, Gly 302 and Ile 301 (Figure 5D). The ZINC32573386 compound was identified as a promising candidate during a screening of the ZINC database. Its predicted docking score was favorable, and the predicted interactions with the active residues of the target protein, PqsA, were also favorable. This suggests that the ZINC32573386 compound could be a potential lead for further drug development and testing. This compound was observed to form three polar and three hydrophobic contacts with the active site residues Gly 269, Thr 304, Ile 301, Gly 279 and Pro 281 and fit well into the active site pocket of the enzyme (Figure 5A). The ZINC79107864 demonstrated a docking score of −7.87 and had favorable interactions with the key areas of the target protein. The way in which this compound binds revealed that it established six polar connections with the active site amino acids Phe 209, Gly 279, Ile 301, and Thr 304, as seen in Figure 5B. The Phe 209 and Pro 281 residues interacted with the aromatic ring of the lead compound, contributing to its effectiveness as a strong inhibitor due to electron-donating and electron-withdrawing groups and delocalized electrons on the aromatic ring. Table 1 presents the outcomes for the finally selected lead compounds.

### 3.4. Molecular Dynamics Simulation

In order to obtain dynamical information under explicit solvent conditions, all four complexes (ZINC32573386, ZINC79107864, ChemBridge53910279, and ChemBridge54245649) along with the reference complex were subjected to all-atom MD simulations for 100 ns. The Amber 22 program was utilized to perform a 100 ns molecular dynamics simulation (MDS) in order to identify the structure of the lead compound complexes that were both well-stabilized and equilibrated. The stabilities of the four selected complexes as well as the reference complex were determined by calculating the root mean square deviation (RMSD) of the backbone atoms. It was found that there was an inverse relationship between the fluctuation amplitudes of the calcium atoms and the stability of the system. The lower the RMSD, the more stable the system was, and the fewer fluctuations in the c-alpha atoms there were [48]. The ZINC32573386/PqsA complex reached equilibrium during the 100 ns simulation. The RMSD graph revealed that no major fluctuations were recorded during the simulation time, with an RMSD score of 1.1. The RMSD score decreased to 0.9 with negligible changes after 70 to 80 ns. As seen in Figure 6, the system was totally stabilized and achieved an average RMSD value of 1.0 with the fewest variations.

In order to obtain more information on how the inhibition quality of these compounds compared among themselves and with the reference compound and how the retrieved lead compounds made proteins less reactive, root means square fluctuation (RMSF) analysis was performed. Residue flexibility indexing revealed significant information regarding the binding of two proteins, small molecules, molecular recognition, and bioengineering. We calculated the RMSF for the reference complex and the retrieved lead compound reported in Figure 7. The plot revealed that the replacement of one amino acid residue had a different effect on the flexibility of each compound. The RMSF plot showed that the residues of each compound in the same PqsA pocket had entirely different fluctuations in the course of trajectory. The ZINC32573386 showed unusually less residual flexibility fluctuation among (100–140 and 220–300), because it was found to be particularly stable; smaller fluctuations mean a more stable complex. After analyzing the residues, it was found that they had fluctuations similar to those of the reference compound. Further analysis, including assessing binding energies, docking scores, RMSD and RMSF values, showed that ZINC32573386 appeared to be a more potent inhibitor than the reference complex, even though some regions of the reference compound had less fluctuation (Figure 7A)**.** The inhibitors ZINC79107864 and Ch54245649 were found to have less residual fluctuation throughout the entire 100 ns simulation period as compared to the test compound. Although the reference complex appeared stable, these compounds may be better candidates when other analyses are taken into account (as seen in Figure 7B,D). The inhibitor Ch53910279 also exhibited similar RMSF behavior, with fewer fluctuations of 5.1–5.9 Å in the system, specifically in the region of 180–400 amino acid residues. Other regions had fluctuations similar to those of the reference compound and in some regions had greater fluctuations when compared to the test compound. However, when other analyses are considered, Ch53910279 may still be a better candidate than the reference compound (as seen in Figure 7C).

### 3.5. Hydrogen Bond Analysis

To gain a more accurate assessment at the atomic level, the number of hydrogen bonds generated in all systems was calculated. To form a hydrogen bond, two conditions must be met: (1) the hydrogen donor–acceptor angle must be 30 degrees, and (2) the donor–acceptor distance must be 0.35 nanometers. Hydrogen bonds play a crucial role in maintaining the secondary structure of ligands and proteins [49,50]. The time-dependent hydrogen bonding investigation revealed that all of the retrieved lead compounds had the strongest and best hydrogen-bonding networks with the PqsA protein (Figure 8). This revealed that each complex system contained a large number of hydrogen bonds as compared to the reference complex. Our findings show that the generated lead compounds have a high binding ability, making them potentially strong inhibitors of PqsA.

### 3.6. Exploring the Dominant Motions for Lead Compounds

The thermodynamics of binding between ligands and PqsA protein was explored through motion mode analysis. PCA was applied to the coordinate covariance matrix derived from the 100 ns MD simulation of lead compound complexes, allowing the PCs to depict the modes of motion in these complexes. The most prominent motion was depicted by the first PC, but the actual movement of the complexes during simulation was a combination of all PCs. Here, we performed the principal component analysis (PCA) for the ref/PqsA and the ZINC32573386/PqsA, ZINC79107864/PqsA, Ch53910279/PqsA, and Ch54245649/PqsA systems. This could identify the best inhibitor among the finally selected lead compounds as compared to the reference compound. The color gradient (red to green) in the representation highlights the periodic changes in conformation. The PCA plot showed that the lead compounds were compact and stable. Each complex had different motion patterns. The reference compound’s motion was mixed and cluster-like (Figure 9A), with blue dots at the start, followed by red, then blue dots. In contrast, the dots of the first compound ZINC32573386 were more organized and compact compared to the reference compound (Figure 9B), covering a range of −350 to +350 along PC1 and −400 to +350 along PC2. The second inhibitor, ZINC79107864, had a little less assembly in the dots, but they were still more ordered than those of the reference compound (Figure 9C), covering a range of −330 to +380 at PC1 and −380 to +380 at PC2. The third inhibitor, Ch53910279, showed much more dispersion compared to the reference compound (Figure 9D), with reference dots being more compact. The fourth inhibitor, Ch54245649, had less assembly in the dots than the reference compound (Figure 9E), but they were still more organized, covering a range of −330 to +380 at PC1 and −380 to +380 at PC2.

The use of FEL analysis assisted us in determining the stable, native, and meta-stable states of all PqsA systems. In-depth examination of the FEL plot was performed to obtain structural coordinates from the lowest energy states (located at the center of each plot) and comprehend the inhibition mode of both reference and lead compounds, as well as the stability of the overall protein ensemble (Figure 10). In Figure 10C, the blue color in the center signifies the lowest Gibbs energy states. The results from the FEL analysis demonstrate that the reference complex had a single (meta) energy state, while the lead compound had two states (meta-stable and native) separated by low-energy barriers. The stability of the retrieved hits was monitored using Cα-RMSd, and it was noted that they remained stable throughout the MD simulation. On the other hand, the reference compound showed deviation initially but eventually stabilized. The FEL plots of the reference compound showed significant conformational changes. This analysis further demonstrates that the selected hits had the lowest energy barrier throughout the 100 ns trajectory. These findings indicate that the energy states, separated by low, moderate, and high energy barriers, periodically switched from one state to another, causing a periodic change in the conformational behavior.

### 3.7. The Dynamic Cross-Correlation Matrix (DCCM)

In this study, the method of determining functional residue movements in complex proteins, known as the dynamic cross-correlation matrix (DCCM), was employed. The DCCM algorithm was applied to gather insights into the correlated motions of the lead compounds through molecular dynamics simulations. This was accomplished by calculating the cross-correlations of Cα atoms using the R programming language’s Bio3D package [51]. In this research, all post-molecular dynamics simulation results were obtained using AMBER v2022 software. An evaluation of alterations in the conformation of ligands within the binding pocket of the PqsA protein was performed using a DCCM analysis on the Cα atom backbone during 100 ns molecular dynamics simulations. This analysis aimed to identify fluctuations and correlated motions, as illustrated in Figure 11A–E. We analyzed the correlation of movements in the active site of PqsA enzyme by creating DCCM graphs from a 100 ns MD simulation to study the effect of the chosen lead compounds. Positive correlation indicated strong coordination of motions. On the other hand, amino acids demonstrated anti-correlation in motion if their movements were in opposition to one another. DCCM computes both positive and negative (anti-parallel) correlations. The findings demonstrate that, in comparison to the reference complex, every retrieved hit had a unique pattern of correlated motions. However, all of the retrieved hits showed a significant difference between the (+) and (−) correlation of atomic displacements.

### 3.8. Binding Free Energy Calculation

The binding free energy (ΔGbind) was calculated as the difference in the solvation free energy of the complex and the solvation free energy of the individual components. The non-bonded energy (ΔGnon-bonded) was calculated as the difference in the non-bonded interactions between the complex and the individual components. Molecular mechanics generalized Born surface area (MMGBSA), a well-known technique, was used to accurately predict the binding free energies of all of the complexes. Gibbs free energy controls the binding affinity between interacting molecules [52,53]. The table includes the total binding free energies of all of the retrieved hits along with that of the reference complex, as well as VDWAALS, electrostatic energy, polar solvation energy and other energy terms. These findings strongly support that our retrieved hits had greater potential for inhibition than the reference compound. The binding free energies for all of the systems are provided in Table 2.

## 4. Conclusions

The objective of this study was to perform a virtual screening of compounds from the ChemBridge and ZINC databases, as well as molecular docking and molecular dynamics simulations of selected compounds and a reference ligand (6-fluoroanthraniloyl-AMP), and to estimate their binding interactions with the PqsA enzyme. Four natural compounds, ZINC32573386, ZINC79107864, Ch53910279, and Ch54245649, were found to have strong interactions with the active site of the PqsA enzyme, with binding affinities ranging from −6.2 to −9 kcal/moL. These compounds had improved pharmacophore features compared to the reference compound attached to the PqsA (PDB ID;5OE3) protein. The results suggest that these compounds have the potential to be developed into drugs for the PqsA enzyme, which is crucial for quinolone signaling in *P. aeruginosa*. Further in vitro and in vivo clinical testing could validate the compounds as candidate inhibitors of PqsA.

## Figures and Tables

**Figure 1 biomedicines-11-00961-f001:**
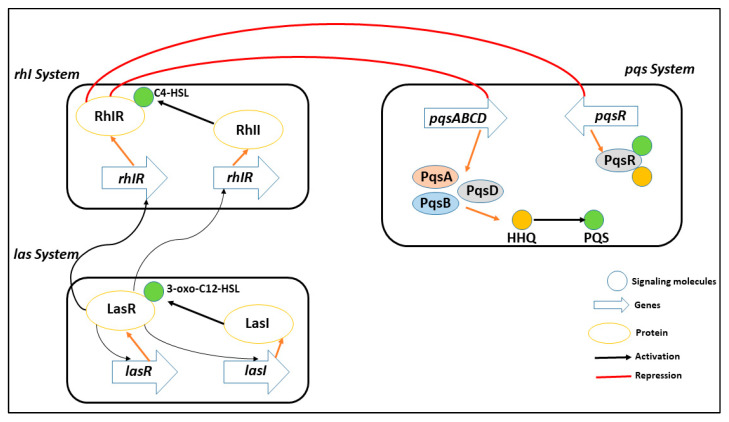
A representative of multiple quorum sensing system in PqsA (rhl, las, and pqs) to coordinate gene expression and adapt to different environmental conditions, including virulence, biofilm formation, and antibiotic resistance. These systems play a crucial role in the pathogenesis of PqsA infection and are potential targets for new antimicrobial strategies.

**Figure 2 biomedicines-11-00961-f002:**
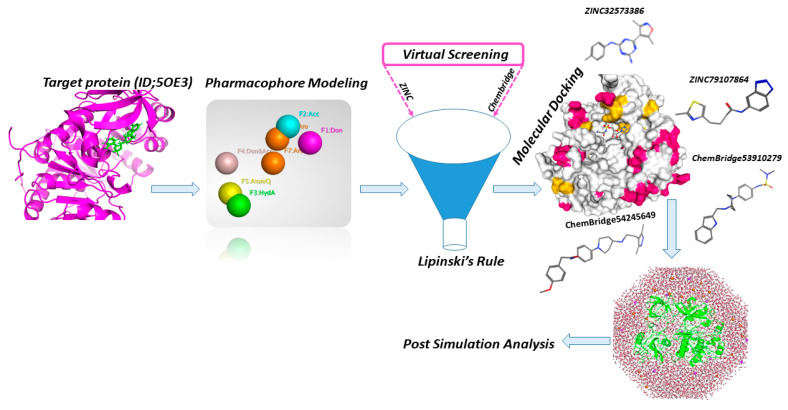
Workflow and tools used in this study for design of potent lead drugs candidates.

**Figure 3 biomedicines-11-00961-f003:**
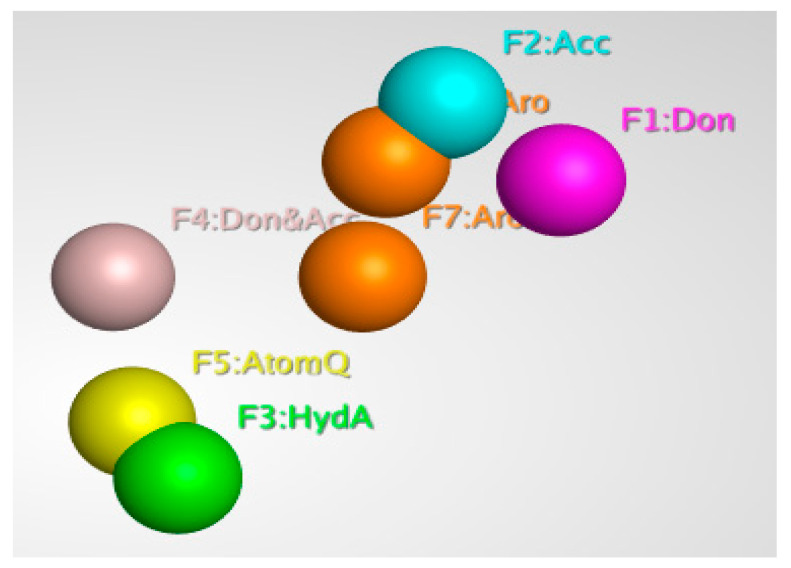
Three-dimensional pharmacophore features of the 6-fluoroanthraniloyl-AMP.

**Figure 4 biomedicines-11-00961-f004:**
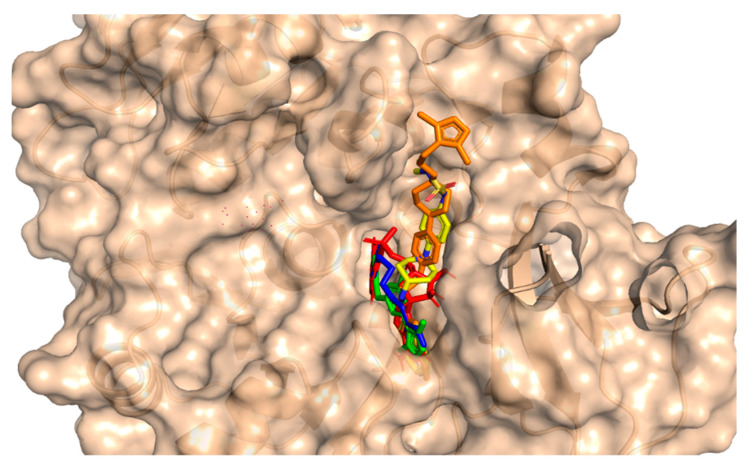
An illustration of the molecular surface of the *Pseudomonas aeruginosa* PqsA enzyme with all active hits superimposed, including the reference compound in red within the binding pocket. The ZINC32573386, ZINC79107864, ChemBridge53910279, and ChemBridge54245649 active ligands were represented by green, blue, yellow, and orange colors, respectively.

**Figure 5 biomedicines-11-00961-f005:**
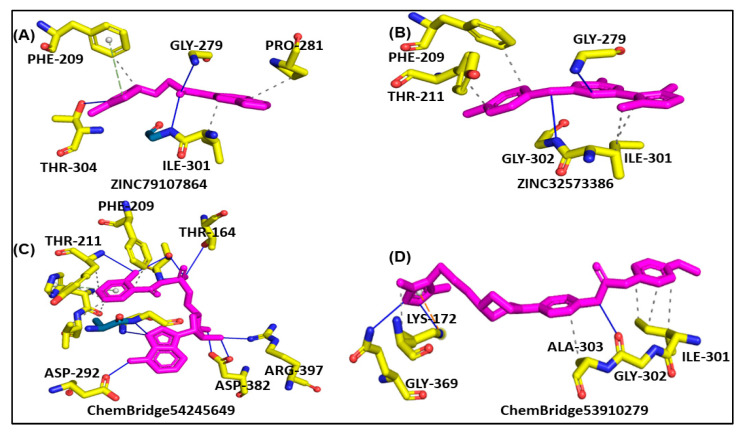
The illustration represents lead compounds attached to the active site of the PqsA enzyme. (**A**,**B**); The binding modes of the inhibitors ZINC79107864 and ZINC32573386, respectively. (**C**,**D**); Binding interaction of ChemBridge54245649 and ChemBridge53910279, respectively.

**Figure 6 biomedicines-11-00961-f006:**
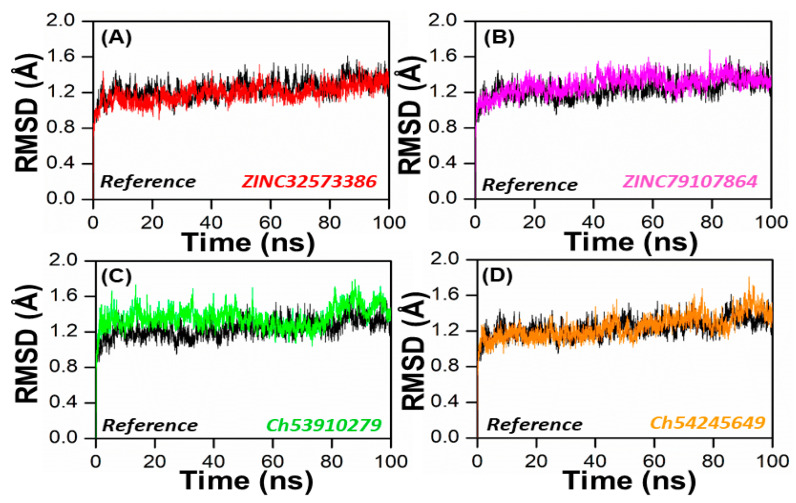
RMSD of the top four hits and the standard compound. (**A**) ZINC32573386/Reference; (**B**) ZINC79107864/Reference; (**C**) Ch53910279/Reference; (**D**) Ch54245649/Reference.

**Figure 7 biomedicines-11-00961-f007:**
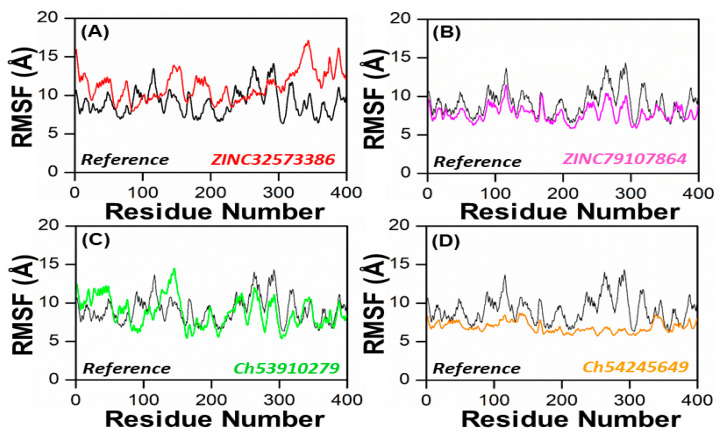
RMSF of the top four hits and the standard compound (reference compound). (**A**) ZINC32573386/Reference; (**B**) ZINC79107864/Reference; (**C**) Ch53910279/Reference; (**D**) Ch54245649/Reference.

**Figure 8 biomedicines-11-00961-f008:**
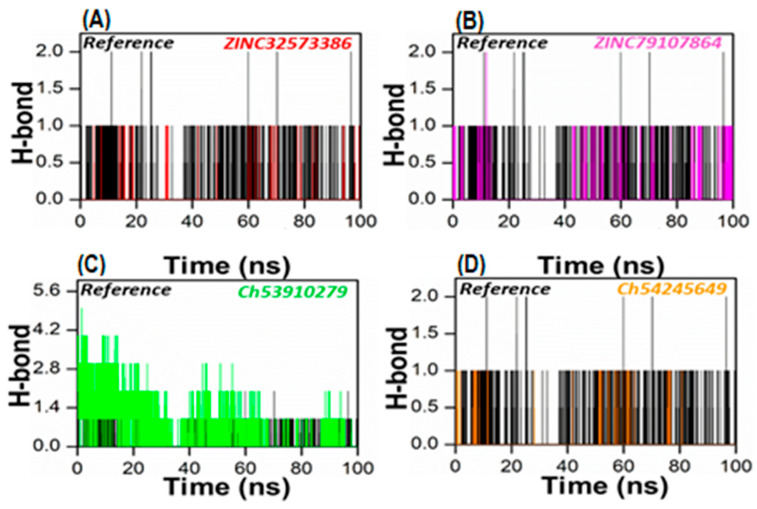
The total number of hydrogen bonds of the reference complex (black), (**A**) ZINC32573386/PqsA (red); (**B**) ZINC79107864/PqsA (magenta); (**C**) Ch53910279/PqsA (green); (**D**) Ch54245649/PqsA (orange). This figure shows the time duration in ns (X-axis) of each system, while the Y-axis represents the number of hydrogen bonds over 100 ns.

**Figure 9 biomedicines-11-00961-f009:**
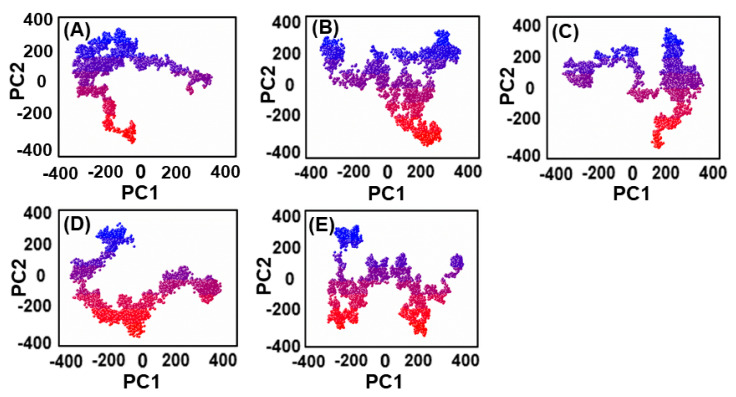
(**A**) The PCA of reference compound; (**B**) ZINC32573386/PqsA; (**C**) ZINC79107864/PqsA; (**D**) Ch53910279/PqsA; (**E**) Ch54245649/PqsA.

**Figure 10 biomedicines-11-00961-f010:**
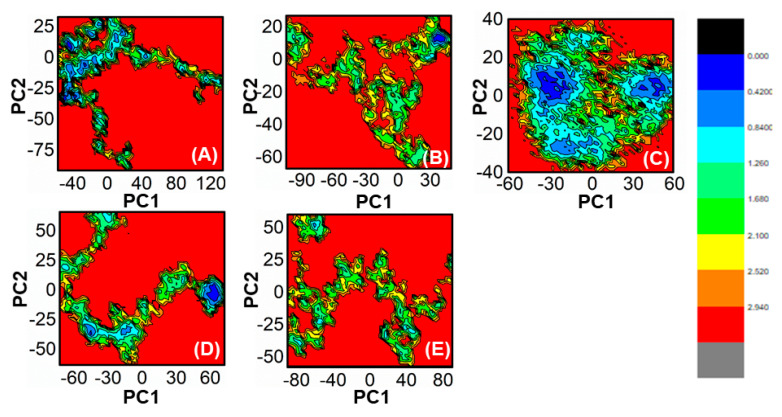
(**A**) The FEL of reference compound; (**B**) ZINC32573386/PqsA; (**C**) ZINC79107864/PqsA; (**D**) Ch53910279/PqsA; (**E**) Ch54245649/PqsA.

**Figure 11 biomedicines-11-00961-f011:**
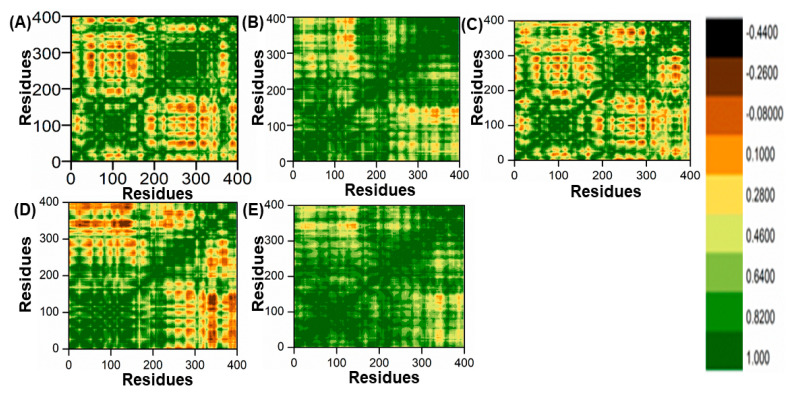
(**A**) DCCM of ref/PqsA; (**B**) ZINC32573386/PqsA; (**C**) ZINC79107864/PqsA; (**D**) Ch53910279/PqsA; (**E**) Ch54245649/PqsA.

**Table 1 biomedicines-11-00961-t001:** The four finally selected lead compounds and their binding energies and affinities.

S.N	ZINC and ChemBridge ID	MW	LogP	Don	Acc	Docking Score	TPSA (Angstrom)	Binding Energy (kcal/moL)
1	ZINC79107864	341.45	3.52	2	4	−7.76	52.93	−52.75
2	ZINC32573386	370.49	3.12	2	5	−8.33	53.96	−43.76
3	ChemBridge54245649	338.43	0.47	3	5	−8.27	44.73	−46.95
4	ChemBridge53910279	366.50	3.45	2	4	−8.90	96.37	−50.48
5	Reference compound	444.21	2.27	4	5	−6.65	143.72	−40.14

**Table 2 biomedicines-11-00961-t002:** Binding free energy calculation.

No	Compound ID	vdW	EEL	ESURF	EGB	TOTAL
1	ZINC32573386	−38.8733	−0.9948	−3.9755	10.3074	−33.5365
2	ZINC79107864	−51.8814	1.0207	−4.6930	10.7405	−44.8132
3	Ch53910279	−46.5828	−43.0180	−5.8361	47.5489	−47.8880
4	Ch54245649	−65.0533	−3.2224	−7.5232	−18.6792	−57.1197
5	Reference	−40.5049	−9.0757	−3.9914	−30.2265	−23.3445

## Data Availability

Not available.

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
