# Peer review of "In Silico Identification of Lead Compounds for Pseudomonas Aeruginosa PqsA Enzyme: Computational Study to Block Biofilm Formation"

_biomedicines, 2023, doi:10.3390/biomedicines11030961_

Round 1

Reviewer 1 Report

The manuscript entitled “In-silico identification of Lead Compounds for Pseudomonas aeruginosa PqsA Enzyme: computational study to Block Biofilm formation” has a significant novelty and written well. The results of the study useful for the discovery of new agents for reduce the bacterial resistance.

The term Pseudomonas aeruginosa must be in italics throughout the manuscript. Abbreviations not defined and not used uniformly (eg., MOE, PBVS, HHQ, QS).Line#53, fluoroquinolones – levo, cipro also same category. Rewrite as other fluoroquinolones or delete levo, cipro from the line. Cite the references for para in lines #74-80.

Author Response

Dear reviewer! please find the attached file.

Reviewer 2 Report

Manuscript biomedicines-2251762 drives the hypothesis of the identification of lead compounds for a Pseudomonas species, that is Pseudomonas aeuruginosa PqsA enzyme with computational thermodynamics data, in order to check if the biofilm formation can be blocked with molecular docking.

Firstly, the manuscript falls within the aims and scope of the journal. It has been prepared according to the guides for authors and is an original contribution, novel to the field. The English language is good. Some corrections, however, must be done. The figures and tables are of good quality. The authors used statistical analysis to find the target compounds and estimated then their binding interactions with the PqsA enzyme. The authors have also used 2 important databases, the ChemBridge and ZINC databases used for the purpose of drug information or development.

In my opinion, the manuscript can be accepted for publication after a revision. I have indicated within the attached pdf the corrections the authors could provide to improve its quality. This paper is the first attempt to understand the quinolone signaling in P. aeruginosa and can benefit the relevant literature, in combination with future studies related to clinical data.

Finally, apart from these future directions the authors mention it is advisable to give the limitations (if any) of this study.

Based on my overall comments, I suggest a minor revision prior to further consideration for publication.

Author Response

(The authors gave the same response as above.)
